# Dopaminergic System in Promoting Recovery from General Anesthesia

**DOI:** 10.3390/brainsci13040538

**Published:** 2023-03-24

**Authors:** Jinxu Wang, Xiaolei Miao, Yi Sun, Sijie Li, Anshi Wu, Changwei Wei

**Affiliations:** Department of Anesthesiology, Beijing Chao-Yang Hospital, Capital Medical University, Beijing 100020, Chinawuanshi1965@163.com (A.W.)

**Keywords:** dopaminergic system, general anesthesia, recovery, neural circuit, dopamine receptor

## Abstract

Dopamine is an important neurotransmitter that plays a biological role by binding to dopamine receptors. The dopaminergic system regulates neural activities, such as reward and punishment, memory, motor control, emotion, and sleep–wake. Numerous studies have confirmed that the dopaminergic system has the function of maintaining wakefulness in the body. In recent years, there has been increasing evidence that the sleep–wake cycle in the brain has similar neurobrain network mechanisms to those associated with the loss and recovery of consciousness induced by general anesthesia. With the continuous development and innovation of neurobiological techniques, the dopaminergic system has now been proved to be involved in the emergence from general anesthesia through the modulation of neuronal activity. This article is an overview of the dopaminergic system and the research progress into its role in wakefulness and general anesthesia recovery. It provides a theoretical basis for interpreting the mechanisms regulating consciousness during general anesthesia.

## 1. Introduction

General anesthesia is a reversible state induced by anesthetics, including loss of consciousness, analgesia, and muscle relaxation [1]. The mechanism of general anesthesia has gone through several parts, including lipid bilayer theory, protein target, and specific receptor target [2,3]. Understanding the mechanisms of reversible alterations of consciousness in the organism during general anesthesia has long been a popular area of neuroscience research. Lately, the mechanism of action of the neural circuit in the regulation of consciousness during general anesthesia has received increasing attention [3,4]. There is increasing evidence that the reversible loss and recovery of consciousness under general anesthesia result from the interaction and modulation of many neural circuits [5]. To ensure surgical safety and advance anesthesia science, it is essential to investigate the mechanisms by which general anesthesia induces reversible changes in consciousness.

Dopaminergic neurons are involved in a wide range of functions in the central nervous system (CNS), including movement, memory, arousal, and cognition [6,7,8,9]. Dopamine (DA) is a major neurotransmitter that exerts biological functions by binding to dopamine receptors [10]. In mammals, the substantia nigra pars compacta (SNc) and the ventral tegmental area (VTA) are areas of the brain rich in dopaminergic neurons, both located in the ventral part of the midbrain [11,12]. Activation of dopaminergic nuclei has been shown to promote arousal, including the VTA, ventral periaqueductal grey matter (vPAG), and dorsal raphe nucleus (DRN) [11,13,14]. The study found that dopaminergic neurons in the VTA and SNc project to prominent pro-arousal brain regions, including the thalamus, locus coeruleus, basal forebrain, and laterodorsal tegmental area [11]. These projections suggest that dopaminergic neurons in the VTA and SNc are intimately involved in maintaining wakefulness. As the brain’s arousal system is inhibited in general anesthesia, and the neural activity of general anesthesia resembles that of sleep, it is suggested that reversible loss and recovery of consciousness caused by general anesthesia may have a similar mechanism to the activity changes of sleep–wake-related nuclei [15,16]. In this review, we provide an overview of dopaminergic neurons, the neural projections they are involved in, and their role in wakefulness and general anesthesia recovery.

## 2. Dopaminergic System

### 2.1. Dopaminergic Neurons

DA is the most abundant monoamine neurotransmitter in CNS and regulates a diverse range of neural activities in the brain. Dopaminergic neurons catalyze tyrosine uptake into dopa in the cytoplasm via tyrosine hydroxylase (TH) [17]. Dopamine decarboxylase (DDC) then acts on this tyrosine to produce dopamine, which is stored in vesicles [18]. Subsequently, dopaminergic neurons release the vesicles into the synaptic terminal, where DA in the synaptic cleft acts on dopamine receptors in the postsynaptic membrane to exert neurotransmitter effects [19,20]. Dopaminergic neurons are anatomically and functionally heterogeneous cell populations widely distributed in the diencephalon, midbrain, and olfactory bulb. Although dopaminergic neurons make up only about 1% of the total number of neurons in the brain, they play an essential role in the regulation of the basic functions of the brain. Most dopaminergic neuron populations are located in the ventral part of the midbrain. They account for approximately 90% of all dopaminergic neurons in the brain, with various physiological functions to accommodate motor regulation and mental activity [21]. Dopaminergic neurons are engaged in three main types of physiological activity: first, the coordination of extrapyramidal movements, including involvement in the pathogenesis of Parkinson’s disease [22]; second, the regulation of mental activity, including functions, such as learning, memory, cognition, mood, depression and the sleep–wake cycle [9,23,24,25]; and third, the modulation of endocrine system functions, such as tuberoinfundibular dopaminergic neurons that regulate the secretion of prolactin [26].

### 2.2. Dopaminergic Neuron Projections

The dopaminergic nervous system is divided into the ascending dopaminergic neural projection system, the descending dopaminergic neural projection system, and the local dopaminergic projection system in the aqueduct of Sylvius and the periventricular region [27]. Dopaminergic neurons in the brain are concentrated in the SNc, the VTA, the hypothalamus, and the periventricular nucleus. Projections from midbrain dopaminergic neurons are received and modulated by numerous brain regions. There are four main dopaminergic pathways in the mammalian CNS: the substantia nigra–striatal pathway, the midbrain–limbic system pathway, the mesolimbic–cortical pathway, and the tubero–infundibular pathway. The substantia nigra–striatal pathway, the dopaminergic pathway from the SNc to the striatum, is a crucial component of the basal ganglia, which is primarily implicated in the regulation of movement and whose deterioration causes Parkinson’s syndrome [28]. The midbrain–limbic system pathway, which connects the VTA to the nucleus accumbens (NAc), hippocampus, and amygdala, is associated with reward, reinforcement, and emotion [29]. On the other hand, the mesolimbic–cortical pathway is a projection of cells from the medial VTA to the prefrontal, cingulate, and peripheral cortices, complementing the function of the mesolimbic pathway and contributing to cognition [30]. There is a considerable degree of overlap of dopaminergic neurons between VTA neurons projecting to different targets. Therefore, the two systems involved in the VTA are usually referred to as the mesocortical–limbic system [31]. The nodal funnel bundle arises from the hypothalamus’s arcuate nucleus, projects to the hypothalamus’s midline, and regulates prolactin release [32]. In addition to the four main pathways mentioned above, there is another mesolimbic dopaminergic cluster in the posterior dorsal hypothalamus called area A11 that is involved in perception [33]. There are also some short-range dopamine projections, such as DA amacrine cells (DACs) with long axon-like synapses that extend to the inner nuclear layer, ganglion cell layer, and sometimes to the outer plexiform layer, which overlap and branch to form a dense dendritic network [34]. Depending on their distance and dense network, DACs have the potential to influence activity at different levels in the retina [35].

### 2.3. Dopamine Receptors

Dopamine receptors (DRs) belong to the G protein-coupled receptor family, have a typical seven-transmembrane structure, signal via a G protein-dependent and independent mechanism, and are widely expressed in the CNS [36]. DA exerts effects by binding to respective membrane receptors. Based on their different binding to ligands and pharmacological properties and signaling pathways, DRs are classified into D1-like and D2-like categories [37]. D1-like DRs contain D1R and D5R, while D2-like DRs contain D2R, D3R, and D4R, of which D1R and D2R are widely distributed in the CNS, whereas the other subtypes are small and restricted in distribution [38,39]. D1-like receptors are found mainly in the caudate nucleus, NAc, substantia nigra pars reticulate, amygdala, frontal cortex, and olfactory bulb [40]. D2-like receptors are mainly expressed in the striatum, VTA, NAc, hypothalamus, hippocampus, and cortex [41]. DA binding to D1-like receptors activates adenylate cyclase, leading to increased cAMP levels and increased cAMP-dependent protein kinase (PKA) activity, resulting in physiological consequences [42]. However, DA binds to D2-like receptors to inhibit adenylate cyclase, decreasing cAMP content [43]. Although cAMP signaling is most relevant to the activation of DRs, some studies have shown that DRs exert biological effects through alternative signaling pathways that are not dependent on cAMP. For example, activation of both D1-like and D2-like receptors can activate protein lipase C-β to induce inositol triphosphate-mediated intracellular calcium flux [44]. Both D1R and D2R trans-activate brain-derived neurotrophic factor receptors in neurons [45]. In addition, DRs regulate the internalization of calcium ions through protein-to-protein interactions [46]. Direct interaction of D1 and D2 receptors with Na+-K+-ATPase has also been demonstrated [47]. Thus, dopamine action depends on the classical adenylate cyclase/cAMP/PKA pathway but is also influenced by second messenger responses, postsynaptic plasma membrane ion channels, and protein expression profiles [36].

## 3. Dopaminergic System and Arousal

The natural sleep–wake regulation system is composed of several neural clusters that form an interactive neural network. Dopaminergic-related neural circuits play a vital role in natural wakefulness [48]. The ascending dopaminergic system’s SNc, VTA, striatum, and pallidum receive innervation from sleep–wake regulatory nuclei, forming an interacting neural network that regulates wake-related behavior [49,50]. Studies have shown that the release of DA in the cerebral cortex is closely aligned with arousal levels [51]. The string firing pattern of dopaminergic neurons may be relevant to the generation of arousal EEG. Dopaminergic neurons in the SNc and VTA evidenced prominent string firing during arousal [52]. The cell membranes of dopamine neurons contain proteins that facilitate dopamine reuptake, known as dopamine transporters. Treatment with the dopamine transporter inhibitor GBR12909 results in a dose-dependent increase in arousal levels [13]. In addition, damage to dopaminergic neurons can induce a coma-like state [53].

There are numerous arousal-promoting nuclei in the CNS, of which dopaminergic projections from the periaqueductal grey (PAG), VTA, and NAc are closely associated with arousal. The PAG receives input from the ascending spinal pathway, amygdala, and hypothalamus and projects to a wide range of brain regions, including the ventral lateral medulla and hypothalamus [54]. The ventral periaqueductal plenum (vPAG) is associated with sleep, and selective chemical damage to dopaminergic neurons in the vPAG significantly increases sleep duration [13]. Chemogenetic activation of dopaminergic neurons of locus coeruleus-vPAG or direct activation of vPAG-DA neurons promotes arousal [55]. The role of vPAG-DA neurons in sleep–wake is associated with their projections to multiple pro-arousal brain regions, including the prefrontal cortex (PFC), ventral lateral preoptic area, and locus coeruleus [13,56,57]. The vPAG-DA neurons coordinate orexinergic, cholinergic, GABAergic, and noradrenergic neurons to impact sleep–wake processes. [13,58]. The VTA is a major concentration of DA neurons, and multiple studies have confirmed its role in promoting arousal. EEG spectral analysis revealed that activation of the VTA-DA to the prelimbic cortex (PrL) pathway during sevoflurane anesthesia decreased sleep-related low-frequency activity and increased wake-related high-frequency activity [59]. In addition, extracellular DA levels in the prefrontal cortex were increased, and arousal was enhanced by injecting orexin-A into the VTA [60]. NAc, a critical brain region for the projection of dopaminergic nuclei, is also implicated in sleep–wake regulation. A recent study showed that optogenetics activation of the neural circuits projected by VTA dopaminergic neurons to NAc promoted arousal in mice [61]. Extracellular DA levels in the NAc decrease during non-rapid eye movement sleep and increase significantly during wakefulness and rapid eye movement sleep [62]. The study found that the DRs of the NAc are involved in regulating the sleep arousal mechanism [63]. Moreover, age-related downregulation of D1R in the NAc shell results in decreased function of D1R in promoting arousal [64]. The ventral pallidum (VP) is a major component of the basal ganglia, which receives intensive input from the NAc and a large number of VTA dopaminergic projections [65,66]. Studies have found that NAc D1R neurons regulate wakefulness, while NAc D2R neurons projecting to the VP regulate sleep [67,68]. It was shown that in vivo stimulation of VP-GABAergic neurons increased VTA-dopaminergic neuronal activity, which in turn induced arousal, whereas pretreatment with dopaminergic antagonists completely blocked arousal induced by VP-GABAergic neuronal activation [65]. This suggests that VTA-dopaminergic neurons mediate arousal through activation elicited by VP-GABAergic neurons.

## 4. Dopaminergic System for General Anesthesia Recovery

General anesthesia is widespread in clinical practice, but the mechanisms by which general anesthesia causes a reversible loss and recovery of consciousness still need to be well elucidated. It is generally accepted that the CNS is interconnected by neuronal projections between nuclei forming a complex and diverse neural network achieved by the interaction of multiple neurotransmitters and neuromodulators [69,70]. Dopaminergic neurons are integral to neural network regulation and perform critical functions in cognition, memory, sleep, and general anesthesia [7,9,11,71]. Identifying the mechanisms and patterns of emergence regulation in anesthesia is central to improving the quality of recovery from general anesthesia. A growing number of studies have documented that central dopaminergic system activation is associated with the emergence of general anesthesia [59,72]. Current research into the dopaminergic system regulating consciousness in general anesthesia has focused on the dopaminergic nuclei, associated neural circuits, and dopamine receptors (Table 1).

### 4.1. Nervous Nuclei and General Anesthesia

#### 4.1.1. VTA

Dopaminergic neurons in the VTA are thought to be essential in altering consciousness induced by general anesthesia. As an integral part of the upstream activation system, projections from the VTA to the basal forebrain, NAc, and prefrontal cortex contribute to cortical and behavioral activation [73]. Bilateral VTA lesions significantly prolong the emergence time in propofol anesthesia but exert no impact on induction time. However, isoflurane and ketamine anesthesia are insensitive to VTA lesions [85]. This may be due to different mechanisms of action between the different anesthetic drugs. Dopamine transporter (DAT) is a transmembrane protein that is specifically distributed on the surface of DA neurons and regulates extracellular dopamine concentrations by transporting extracellular DA into the cell [74]. A recent study revealed that DAT inhibition in the VTA enhances PFC neuronal activity, thereby facilitating emergence in rats under propofol anesthesia [75]. Furthermore, electrical stimulation of VTA-DA neurons induced rapid arousal under isoflurane or propofol anesthesia. Consciousness and EEG change consistent with arousal during general anesthesia were restored with electrical stimulation of the VTA brain region but not observed with SNc stimulation [76]. Optogenetic manipulation of the VTA-DA under isoflurane anesthesia has been reported to reduce the time to wakefulness [77]. Li et al. found that microinjection of orexin-A into the VTA of rats reduced the burst suppression ratio during the maintenance of isoflurane anesthesia and promoted emergence from isoflurane anesthesia. In isolated brain slices, orexin-A also directly increased action potential frequency of dopaminergic neurons in VTA with 1 MAC isoflurane, implying that projections from orexinergic neurons to VTA-DA neurons may be related to the emergence of general anesthesia by affecting the neuronal activity of DA neurons [86]. Selective activation of VTA DA neurons during sevoflurane anesthesia leads to prolonged induction and shortened recovery time associated with sevoflurane anesthesia [62]. These results suggest that VTA-DA neurons are involved in the induction and emergence of general anesthesia and that there are differences in the alterations in consciousness under different anesthetics. Further research is necessary to elucidate the mechanisms of anesthetics in different states of anesthesia.

#### 4.1.2. NAc

The NAc is a significant component of the ventral striatum and is constituted by medium-spiny GABAergic neurons [87]. Medium spiny neurons (MSNs) in the NAc shell receive projections primarily from dopaminergic neurons in the VTA, and the core of the NAc receives projections primarily from the substantia nigra [88]. Recent studies have demonstrated that the dopaminergic system is associated with the modulation of consciousness in the NAc during general anesthesia. Using the microdialysis technique to observe changes in DA concentrations in the NAc of rats under propofol anesthesia, it was found that low doses (9 mg/kg) decreased DA concentrations, while medium doses (60 mg/kg) and high doses (100 mg/kg) increased DA concentrations. This suggests that there may be variations in the activity of DA neurons in NAc at different anesthetic doses. [89]. Gui et al. reported that DA in the NAc diminished with the loss of consciousness induced by sevoflurane anesthesia, remained stable during anesthesia, and increased in the transition from anesthesia to arousal [62]. NAc is a potential target for the involvement of the dopaminergic system in the regulation of consciousness by general anesthesia due to its high expression of D1R and D2R, as well as its complex fiber connections with cortical and midbrain dopaminergic nuclei [78]. Fiber-optic recording of changes in calcium-signaling NAc activity during propofol anesthesia revealed that the neuronal excitation of rat NAc neurons decreased during the induction and restored during anesthesia emergence [72]. Other results using the same fiber-optic calcium-sensing technology showed reduced neuron activity in NAc expressing D1R during induction and maintenance of sevoflurane and enhanced neuron activity during wakefulness. NAc-D1R neurons induce cortical activation when activated by optogenetic stimulation, implicating NAc-D1R neurons as promising targets for the modulation of consciousness [90]. By studying specific subtypes of neurons in more detail, it is possible to identify the specific mechanisms that influence general anesthesia arousal, which also provides a new direction for future mechanistic research.

#### 4.1.3. vPAG

The vPAG-DA is concerned with regulating consciousness by general anesthesia through projections to several nuclei, notably the basal forebrain, locus coeruleus, and ventral lateral preoptic area [91,92,93]; 6-OHDA is a potent dopamine neurotoxicant extensively applied as a selective catecholaminergic neurotoxic agent in cellular or animal models of Parkinson’s disease [79,94]. Damaging dopaminergic neurons in vPAG by 6-OHDA increased EEG δ wave, shortened induction time, and prolonged recovery time of propofol anesthesia, indicating that vPAG-DA neurons are related to the anesthetic mechanism of propofol [80]. In addition, propofol increased the sensitivity of postsynaptic GABAA receptors and the release of the presynaptic inhibitory neurotransmitter GABA in vPAG-DA neurons, ultimately inhibiting the activity of vPAG dopamine neurons. Microinjection of a GABAAR antagonist abolished the suppressive efficacy on vPAG-DA neurons, suggesting that GABAA receptors are engaged in isoflurane anesthesia through the regulation of vPAG-DA [80]. Liu et al. discovered that isoflurane anesthesia caused a decrease in DA neuronal excitation in the vPAG. In contrast, the activity of the whole population of vPAG neurons increased dramatically during the emergence of isoflurane anesthesia. The induction time is shortened, and the recovery time is lengthened by unilateral ablation of vPAG-DA neurons. Furthermore, isoflurane significantly increased presynaptic inhibitory neurotransmitter GABA release from vPAG-DA neurons by increasing spontaneous inhibitory postsynaptic current (sIPSC) frequency and decay time [71].

### 4.2. Neural Circuit and General Anesthesia

The dopaminergic system in general anesthesia not only influences the anesthetic state of the body by altering DA levels in the corresponding nuclei but also participates in the regulation of consciousness through neurocircuitry (Figure 1). Orexin injections into the VTA were found to elevate extracellular DA levels in the PFC and prolong wakefulness [60]. These findings suggest that there are dopaminergic projections in the VTA-PFC and that the projections of orexinergic neurons in the VTA may correlate with levels of DA release in the PFC and are involved in the effects on consciousness. In general anesthesia, this has been well demonstrated. Consciousness modulation by sevoflurane anesthesia is associated with dopaminergic projections from VTA to PrL, part of the PFC. Chemogenetics and optogenetics suggest that the VTA-PrL dopaminergic pathway participates in the mechanism of sevoflurane anesthesia and that activation of this pathway prolongs the induction of anesthesia and enhances anesthesia emergence [59]. Using DA neurotransmitter probes to record changes in DA in the neural nuclei, dexmedetomidine was found to increase DA concentrations in the medial prefrontal cortex and NAc in the dopaminergic neuronal projection area through activation of VTA dopaminergic neurons, which explains the rapid awakening after dexmedetomidine sedation and reflects the vital role of the VTA-NAc/mPFC dopaminergic pathway in promoting awakening during general anesthesia [81]. In addition, Gui et al. found that stimulation of the VTA-NAc dopaminergic pathway extended the induction of sevoflurane anesthesia in mice and at the same time shortened the wake time, accompanied by a decrease in δ waves and an increase in γ waves, whereas inhibition of the pathway had the opposite effect [62]. Activation of orexin neural projection from the periaqueductal lateral hypothalamic area (PeFLH) to the VTA promotes emergence from isoflurane anesthesia by modulation of dopaminergic neurons [86]. The dopaminergic system has been identified as an eminent contributor to the arousal-promoting properties of general anesthesia. As research continues, it is becoming clear that the regulation of consciousness during general anesthesia is achieved through the interaction and co-regulation of multiple types of neurons.

### 4.3. Dopamine Receptor and General Anesthesia

Depending on the type of neuronal terminals they receive and the type of neurons to which they project, dopamine receptor neurons play different roles in regulating arousal in different regions of the brain. Over the past years, dopamine receptors have been implicated in the regulation of consciousness during general anesthesia. Systemic administration of D1R agonists or methyl benzene significantly accelerated recovery in isoflurane-anesthetized rats [82,95]. The D1R agonist choro-APB shortened the time to wakefulness after isoflurane general anesthesia, and the D1R antagonist SCH-23390 inhibited this effect. In contrast, awakening after isoflurane general anesthesia was not induced by the D2R agonist quinpirole [95]. D-amphetamine reversed dexmedetomidine-induced unconsciousness by promoting dopamine release and activating D1 and D5 receptors in the brain. Furthermore, D-amphetamine induced recovery from propofol and sevoflurane anesthesia. This effect is not seen in ketamine anesthesia, implying that ketamine may have a different pharmacological mechanism than other general anesthetics [83,84]. These studies are consistent with the involvement of expressing DR neurons in recovering consciousness after anesthesia.

In the course of the research, DRs have been targeted in specific nuclei, and it has been confirmed that dopamine receptors in various nuclei engage in the mechanism of action of general anesthesia. Optogenetic activation of the VTA-DA under isoflurane anesthesia has been reported to shorten the time to wakefulness, which can be inhibited by intraperitoneal injection of D1R blockers [77]. To accurately determine the role of DRs during general anesthesia, DR agonists or antagonists are used on specific nerve clusters. Both intraperitoneal injection and PrL microinjection of D1R agonist Choro-APB promoted recovery from sevoflurane anesthesia, while SCH23390, a D1R antagonist, enhanced anesthesia, suggesting the D1R in PrL is an essential regulatory site of the dopaminergic system in sevoflurane anesthesia [59]. NAc dopaminergic receptors exert maintaining arousal. Systemic activation of D1R, but not D2R, shortens the recovery time from isoflurane anesthesia, suggesting that D1R mediates arousal in general anesthesia [68,95]. Moreover, microinjection of D1R agonists into the NAc shell accelerated recovery from isoflurane anesthesia in mice, whereas D1R antagonists delayed wakefulness [64]. Zhang et al. found that D1R receptor regulation of MSNs activity of NAc is essential for the recovery of propofol-induced loss of consciousness. D1R is involved in the increased frequency and prolonged decay time of sIPSCs and miniature inhibitory postsynaptic current (mIPSCs) of the MSNs induced by propofol anesthesia [72]. Optogenetics activation of D1R neurons in NAc induces cortical activation and recovery during sevoflurane anesthesia, which is also reflected in burst suppression oscillations induced by deep sevoflurane anesthesia [90]. Aging may attenuate anesthetic arousal by downregulating D1R expression in the NAc shell [64]. Therefore, changes in the dopaminergic system in response to age-related factors, including DA neuronal damage, reduced DA synthesis, and downregulation of DR, all influence the regulation of consciousness by general anesthesia. In addition, the olfactory nodule (OT) is intensively innervated by dopaminergic neurons of VTA, and its main neuronal components are neurons expressing D1R and D2R. Injection of D1R or D2R agonists into the OT significantly accelerates isoflurane-induced wakefulness, whereas D1R or D2R antagonists significantly delay wakefulness [96].

## 5. Conclusions

In summary, DA neurons are involved in the regulation of both wakefulness and recovery from general anesthesia. The dopaminergic system plays a vital role in the regulation of consciousness during general anesthesia. It is involved in the mechanism of action through nuclei, associated neural circuits, and dopamine receptors. Nevertheless, the mechanisms underlying the regulation of consciousness by general anesthetics remain poorly understood. With the development and innovation of scientific techniques, such as optogenetics, chemogenetics, neurotransmitter probes, electroencephalography, and so on, it is possible to explore in greater depth the mechanisms by which general anesthesia regulates consciousness. These techniques allow us to explore the dynamics of neurotransmitters and neuropeptides in general anesthesia and how they are involved in the altered consciousness of general anesthesia, the nuclei and their interactions in general anesthesia, and even provide a clearer understanding of the specific pharmacological effects of different general anesthetics. Dopaminergic neurons are involved in facilitating recovery of consciousness from general anesthesia, and it is expected that improvements in the clinical problem of delayed awakening will be achieved. In this review, dopaminergic neurons and their role in arousal and recovery under general anesthesia are summarized, providing a theoretical basis for the pharmacological mechanism of general anesthetics in regulating consciousness.

## Figures and Tables

**Figure 1 brainsci-13-00538-f001:**
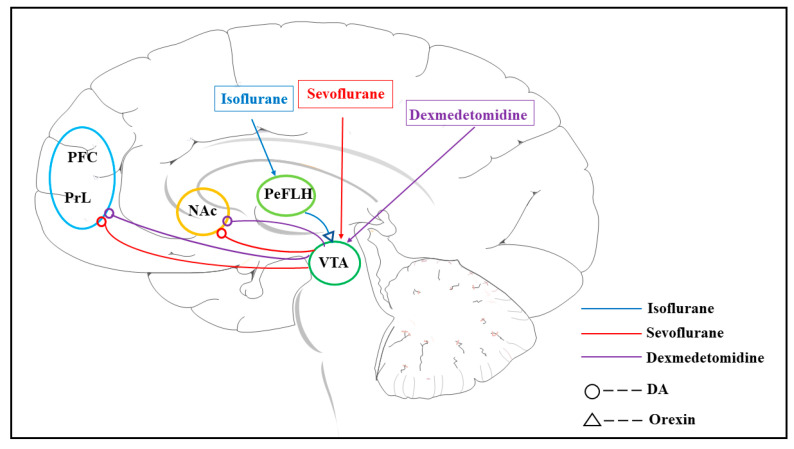
Neural circuit of the dopaminergic system in general anesthesia. Neural circuits are involved in the dopamine system under different anesthetics on recovery from general anesthesia. Circles represent dopaminergic neural projections, and the triangle represents orexin neural projections. The arrows indicate that the anesthetics act on the corresponding nucleus, and the colors are used to identify the different anesthetics. The neural circuits under the same general anesthetic have the same color line; abbreviations: the prefrontal cortex (PFC), prelimbic cortex (PrL), nucleus accumbens (NAc), periaqueductal lateral hypothalamic area (PeFLH), ventral tegmental area (VTA), Dexmedetomidine (Dex), and dopamine (DA).

**Table 1 brainsci-13-00538-t001:** The dopaminergic system is involved in regulating consciousness during general anesthesia; abbreviations: Sprague Dawley (SD), the ventral tegmental area (VTA), dopamine (DA), the dopamine receptor (DR), substantia nigra (SN), the dopamine transporter (DAT), nucleus accumbens (NAc), medium spiny neurons (MSNs), ventral periaqueductal grey matter (vPAG), 6-hydroxydopamine (6-OHDA), the prelimbic cortex (PrL), the medial prefrontal cortex (mPFC), dexmedetomidine (Dex), the periaqueductal lateral hypothalamic area (PeFLH), tyrosine hydroxylase (TH), and the olfactory tubercle (OT).

Animal	Nervous Nuclei/ Neural Circuit	General Anesthetics	Intervention /Methods	Findings in Consciousness during General Anesthesia	Reference
SD rats	VTA-PrL	Sevoflurane	D1R agonist and antagonist were intraperitoneally injected or microinjected into PrL. Chemogenetics and Optogenetics	The dopaminergic pathway of VTA-PrL promotes the emergence of sevoflurane anesthesia.	[58]
DAT-cre mice	VTA, VTA-NAc	Sevoflurane, isoflurane	Chemogenetics, Optogenetics, and DA sensor based genetically encoded	Activation of the VTA-NAc dopaminergic pathway delayed the induction and promoted the emergence from general anesthesia	[61]
SD rats	vPAG	Isoflurane	Microinjection of 6-OHDA into vPAG. Whole-cell patch clamp recording	Lesion to the vPAG-DA neurons shortens the induction time and prolongs the recovery time in isoflurane anesthesia by activating the GABAA receptor	[70]
SD rats	NAc	Propofol	Microinjection of DR agonists and antagonists into NAc. Whole-cell patch clamp recording	The D1R of the NAc MSNs is involved in regulating the emergence of propofol-induced unconsciousness.	[71]
SD rats	VTA	Isoflurane, propofol, ketamine	6-OHDA lesioned Bilateral VTA-DA neurons	VTA-DA neurons may be involved in the emergence of propofol	[73]
SD rats	VTA	Propofol	Knockdown DAT	DAT inhibition in VTA enhances PFC neurons activity and promotes recovery after propofol anesthesia.	[74]
SD rats	VTA, SN	Isoflurane, propofol	Electrostimulation	Electrical stimulation of VTA, but not SN, induced recovery from anesthesia	[75]
DAT-cre mice	VTA	Isoflurane	Optogenetics activation of VTA-DA. DR antagonists were used to intervening	Selective stimulation of VTA-DA neurons induced emergence from general anesthesia	[76]
Hcrt-cre rats	VTA PeFLH- VTA	Isoflurane	Microinjection of Orexins into VTA. Identification of Orexin receptors and DA neurons in VTA by immunofluorescence. Optogenetics	Orexin promotes the emergence of isoflurane anesthesia by activating DA neurons in VTA.	[77]
D1R-cre mice	NAc	Sevoflurane	Chemogenetics and Optogenetics	Activation of NAc D1R neurons induced cortical activation and behavioral emergence during sevoflurane anesthesia	[78]
SD rats	vPAG	Propofol	Microinjection of 6-OHDA into vPAG. Whole-cell patch clamp recording	Lesion to the vPAG-DA neurons shortens the induction time and prolongs the recovery time during propofol anesthesia	[79]
DAT-cre mice	VTA-NAc, VTA-mPFC	Dex	DA sensor based genetically encoded. Chemogenomics	Dex activates DA neurons in the VTA and increases DA concentration in the NAc and mPFC.	[80]
SD rats	/	Isoflurane	Intraperitoneal injection of DR agonists and antagonists	Activation of D1R mediates emergence from isoflurane	[81]
SD rats	/	Sevoflurane, propofol	Dextroamphetamine or atomoxetine is injected intraperitoneally during anesthesia.	Dextroamphetamine induced recovery from general anesthesia, whereas atomoxetine did not	[82]
SD rats	/	Dex, ketamine	Dextroamphetamine is administered intravenously during anesthesia	Dextroamphetamine induced recovery after anesthesia with Dex via D1R or D5R, but not with ketamine	[83]
TH-IRES-cre mice	OT	Isoflurane	Microinjection of DR agonists and antagonists into OT. Optogenetics	The dopaminergic pathway in OT accelerated emergence from isoflurane anesthesia	[84]

## Data Availability

No applicable.

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
