# Peer review of "Dopaminergic System in Promoting Recovery from General Anesthesia"

_brainsci, 2023, doi:10.3390/brainsci13040538_

Round 1

Reviewer 1 Report

Thank you for permitting me to review this manuscript 

Line 48 PPR (please provide reference)

Line 107-108 please provide additional explanationations or examples 

 Table  please separate references with molecules promoting sleep , and those promoting emergence from anesthesia and avoid a meltdown 

The authors should provide a  simple summary to permit the reader to  have  a message to retain.  

figure 1 please add dexmedetomidine totally or precise in the legend in order not to confuse with dextroamphetamine 

In addition this figures needs  more legend and some more explanation , if this is about recovery of general anesthesia , please as for sevoflurane , isoflurane and dex , which acts on VTA , whats is the role and the circuit for PEFLH ? 

Line 190 , please elaborate , which state of anesthesia ?

Line 209 please precise  that  the dosage  which displayed are for rats , since  for humans dosage of propofol range 2 to 3 mg/kg  therefore   may be extrapolation to humans in this case is difficult 

Author Response

We thank Reviewer #1 for the comments and suggestions regarding our manuscript. According to your comments, we have revised the manuscript accordingly. We hope that these modifications meet your satisfaction.

  1. Line 48 PPR (please provide reference)

Response: We appreciate this excellent suggestion. Relevant references have been added in line 48, and mark them in red.

  1. Line 107-108 please provide additional explanationations or examples 

Response: We appreciate this excellent suggestion. We have added examples to illustrate this section in lines 117-125.

  1. Table please separate references with molecules promoting sleep, and those promoting emergence from anesthesia and avoid a meltdown 

Response: We appreciate this excellent suggestion. The table reflects the role of dopaminergic neurons in promoting recovery from general anesthesia and we have modified it to avoid confusion.

  1. The authors should provide a simple summary to permit the reader to have a message to retain.  

Response: We appreciate this excellent suggestion. We have provided a “Graphical Abstract”, and summarized the contents of this review in the “Conclusion

  1. figure 1 please add dexmedetomidine totally or precise in the legend in order not to confuse with dextroamphetamine 

Response: We appreciate this excellent suggestion. We have shown the full name of dexmedetomidine in Figure 1.

  1. In addition this figures needs more legend and some more explanation , if this is about recovery of general anesthesia , please as for sevoflurane , isoflurane and dex , which acts on VTA , whats is the role and the circuit for PEFLH ? 

Response: We appreciate this excellent suggestion. We have supplemented the legend of the figure and labeled the neural circuits corresponding to the anesthetics in the Figure.

  1. Line 190, please elaborate, which state of anesthesia?

Response: We appreciate this excellent suggestion. We have revised the state of anesthesia in lines 218-222.

  1. Line 209 please precise that the dosage  which displayed are for rats , since  for humans dosage of propofol range 2 to 3 mg/kg  therefore   may be extrapolation to humans in this case is difficult 

Response: We appreciate this excellent suggestion. According to the reference[68], the dosage of propofol is for rats and not for humans.

Reviewer 2 Report

The review of  Jinxu Wang et al. entitled Dopaminergic system in promoting recovery from general anesthesia” provides a comprehensice summary about role of the DAergic system in the general anesthesia, however, I would have some minor recommendations to further improve the quality of the manuscript.

Content:

1. Line 76: It is not easy to decide, how many DAergic systems/pathways there are, some experts cathegorize the ulra-short DAergic pathways (e.g. in the amacrine cells of the retina, etc.) as the fifth system.

2. Generally, the ventral pallidum (VP) of the basal forbrain should be at least mentioned in the review, since it is also one of the main targets of DAergic neurons of VTA, and has reciprocal connections both with VTA and NAC, additionally it is involved in regulation of the arousal through anterior thalamic nuclei as well.

3. Possible role of other neuropeptides in addition to orexine should be mentioned. What about possible role of opioids and endocannabinoids?

4. Mood and depression should be also listed at least in lines 66-69, or even more detailed.

5. Fig. 1.: it should be added into the explanation, what does „/” mean.

Formation and style:

6. There is an unnecessary line break between lines 27-28.

7. Line 74: a word seems to be missing after „periventricular” (nucleus?)

8. The font sometimes unnecessary changes, e.g lines 134-137, 166-167, 207-212, 280-282.

Author Response

We thank Reviewer #2 for the comments and suggestions regarding our manuscript. According to your comments, we have revised the manuscript accordingly. We hope that these modifications meet your satisfaction.

  1. Line 76: It is not easy to decide, how many DAergic systems/pathways there are, some experts cathegorize the ulra-short DAergic pathways (e.g. in the amacrine cells of the retina, etc.) as the fifth system.

Response: We appreciate this excellent suggestion. We modified this and added dopamine projection in the retina in lines 94-101.

  1. Generally, the ventral pallidum (VP) of the basal forbrain should be at least mentioned in the review, since it is also one of the main targets of DAergic neurons of VTA, and has reciprocal connections both with VTA and NAC, additionally it is involved in regulation of the arousal through anterior thalamic nuclei as well.

Response: We appreciate this excellent suggestion. We have supplemented the ventral pallidus in regulation of the arousal in lines 169-177.

  1. Possible role of other neuropeptides in addition to orexine should be mentioned. What about possible role of opioids and endocannabinoids?

Response: We appreciate this excellent suggestion. We reviewed the relevant literature and found that most of the studies on the effects of neuropeptides in the dopaminergic system on regulating consciousness in general anaesthesia focused on the orexin, and the opioids and endogenous cannabinoids mentioned by the reviewer need to be explored in more studies, which is also a valuable direction for research.

  1. Mood and depression should be also listed at least in lines 66-69, or even more detailed.

Response: We appreciate this excellent suggestion. We have added the role of dopaminergic neurons involved in mood and depression and described this part in more detail in lines 67-72.

  1. Fig. 1.: it should be added into the explanation, what does „/” mean. Formation and style:

Response: We appreciate this excellent suggestion. We have labelled the figures and explained them accordingly in the figure legend.

  1. There is an unnecessary line break between lines 27-28.

Response: We appreciate this excellent suggestion. We have modified this part accordingly.

  1. Line 74: a word seems to be missing after „periventricular” (nucleus?)

Response: We appreciate the reviewer’s careful reading. We have added “nucleus” after “periventricular”.

  1. The font sometimes unnecessary changes, e.g lines 134-137, 166-167, 207-212, 280-282.

Response: We appreciate the reviewer’s careful reading. We have made the font consistent.

Reviewer 3 Report

The MS "Dopaminergic system in promoting recovery from general anesthesia"

is well written and really interesting. Moreover, the MS is well organized and clear. 

I would suggest minor comments. 

Discussion: I would suggest to add few lines regarding the perspectives for the field (from what you presented where we should go, what can we put in place, how should we adapt our techniques for the future), and what we should do to promote the recovery.

References: A large amount of citations are old, I would suggest to update this section with more recent references. 

Author Response

We thank Reviewer #3 for the comments and suggestions regarding our manuscript. According to your comments, we have revised the manuscript accordingly. We hope that these modifications meet your satisfaction.

  1. Discussion: I would suggest to add few lines regarding the perspectives for the field (from what you presented where we should go, what can we put in place, how should we adapt our techniques for the future), and what we should do to promote the recovery.

Response: We appreciate this excellent suggestion. We add to the perspectives of the dopaminergic system in promoting recovery from general anesthesia and provide potential directions for research in the “Conclusion”.

  1. References: A large amount of citations are old, I would suggest to update this section with more recent references. 

Response: We appreciate this excellent suggestion. We have updated and cited more recent references, and marked them in red.

Round 2

Reviewer 1 Report

The authors have adequately reponded to my queries 

Author Response

We thank reviewer 1 for approval of our response. We are grateful for your review!